# Lactic acid containing polymers produced in engineered *Sinorhizobium meliloti* and *Pseudomonas putida*

**Tam T. Tran**[¤], **Trevor C. Charles**[iD]*

Department of Biology, University of Waterloo, Waterloo, ON, Canada

¤ Current address: London Research and Development Centre, Agriculture & Agri-Food Canada, London, ON, Canada
* tcharles@uwaterloo.ca

**Data Availability Statement:** All relevant data are within the manuscript.

**Funding:** This work was financially supported by a New Directions grant from Ontario Ministry of Agriculture, Food, and Rural Affairs (award number

## Abstract

This study demonstrates that novel polymer production can be achieved by introducing pTAM, a broad-host-range plasmid expressing codon-optimized genes encoding *Clostridium propionicum* propionate CoA transferase (Pct$_{Cp}$, Pct532) and a modified *Pseudomonas* sp. MBEL 6–19 polyhydroxyalkanoate (PHA) synthase 1 (PhaC1$_{Ps6-19}$, PhaC1400), into *phaC* mutant strains of the native polymer producers *Sinorhizobium meliloti* and *Pseudomonas putida*. Both phenotypic analysis and gas chromatography analysis indicated the synthesis and accumulation of biopolymers in *S. meliloti* and *P. putida* strains. Expression in *S. meliloti* resulted in the production of PLA homopolymer up to 3.2% dried cell weight (DCW). The quaterpolymer P (3HB-*co*-LA-*co*-3HHx-*co*-3HO) was produced by expression in *P. putida*. The *P. putida phaC* mutant strain produced this type of polymer the most efficiently with polymer content of 42% DCW when cultured in defined media with the addition of sodium octanoate. This is the first report, to our knowledge, of the production of a range of different biopolymers using the same plasmid-based system in different backgrounds. In addition, it is the first time that the novel polymer (P(3HB-*co*-LA-*co*-3HHx-*co*-3HO)), has been reported being produced in bacteria.

## Introduction

Traditional plastics made from non-renewable fossil fuels have posed a threat to environment and human health. New materials, which are environment-friendly and easily biodegradable, are being searched to substitute for the traditional plastics. Those materials, such as polyhydroxylalkanoate (PHA), polylactic acid (PLA), polyglycolic acid (PGA), and blends of these polymers, can be produced through either the chemical process or biological process [1,2]. However, the chemical process has some drawbacks such as high temperature, long reaction time, uncontrollable chain length or monomer restriction [3,4]. In addition, chemical synthesis results in harmful left-over chemical residues of metal catalysts for the reaction [5]. Metal catalysts used in ring-opening polymerization stage are heavy metals, such as Cu-based catalyst

381646-09), by a Strategic Projects grant from the Natural Sciences and Engineering Research Council of Canada (award number ND2012-1679), by a Discovery grant from the Natural Sciences and Engineering Research Council of Canada (award number 155385) and by Genome Canada, through the Applied Genomics Research in Bioproducts or Crops (ABC) program for the grant titled "Microbial Genomics for Biofuels and Co-Products from Biorefining Processes.

**Competing interests:** The authors have declared that no competing interests exist.

({2-[1-(2,6-diethylphenylimino)ethyl]phenoxy}2Cu(II)) [6]. Therefore, bioprocess has been considered as an alternative way to produce novel polymers.

PHAs are able to be produced naturally in bacteria such as *Cupriavidus necator* H16 (formerly, *Ralstonia eutropha*) [7] [8], *Alcaligenes latus* [9], *Pseudomonas* [10] [11] and the rhizobia [12]. However, PLA and its derivatives have only been produced through genetic engineering [4,5,13–16], by introduction and expression of modified genes for two enzymes, propionate CoA-transferase (e.g. *pct532*) from *Clostridium propionicum* and PHA synthase (e.g. *phaC1400*) from *Pseudomonas*. These studies, including recent reports of PLA/PHA copolymer production [13,14] have mainly focused on engineering *Escherichia coli* strains that are common model systems in the metabolic engineering field. *E. coli* strains have been engineered to produce a broad range of bioproducts such as biopolymers, biofuels, amino acids, and organic acids because molecular tools are intensively studied and widely available. However, *E. coli* strains are not native polymer producers and hence lack the metabolic pathways for the production of usable polymer precursors. Also, it was shown that *E. coli* strains suffer from the stress of polymer production, and they produce side products that might decrease the overall yield of target products. For these above reasons, we focused our study on engineering two representative native polymer producers, *S. meliloti* which produces short-chain-length (SCL) PHA, and *P. putida* which produces medium-chain-length (MCL) PHA, for efficient novel polymer production systems [12,17]. Both organisms are preferred choices for engineering PHA polymer production because of their broad growth substrate capability, relatively fast growth and fully annotated genomes.

Depending on PHA synthase as well as substrate availability, a variety of different polymer types can be achieved, such as homo-polymer (P(3HB), 3-hydroxybutyrate), co-polymer (P (3HB-*co*-3HV, 3-hydroxybutyrate-*co*-3-hydroxyvalerate), ter-polymer (P(LA-*co*-3HB-*co*-3HV, lactic- acid-*co*-3-hydroxybutyrate-*co*-3-hydroxyvalerate), P (3HB-*co*-3HV-*co*-3HHx, 3-hydroxybutyrate-*co*-3-hydroxyvalerate-*co*-hydroxyhexanoate), P (LA-*co*-3HB-*co*-3HP, lactic acid-*co*-3-hydroxybutyrate-*co*-3-hydroxypentanoate) [18–21]. Based on the type, content, monomer composition, and molecular weight distribution, different polymers possess diverse properties allowing a broader spectrum of application. P (3HB) homopolymer exhibits undesirably high melting temperature as well as high crystallinity, resulting in hard and brittle properties [22]. Meanwhile, MCL PHAs (6–14 carbons) are described as having low melting point and high elasticity [23]. Since the melting temperature of MCL PHAs is relatively low, varying from 39ºC to 61ºC, materials made from MCL PHAs are not rigid enough and tend to lose their coherence at rather low temperature [24]. Incorporating a small amount of MCL PHAs into P (3HB) showed some improved properties of the target polymers. Their properties are similar to polyethylene, which is relatively tough and ductile. The crystallinity and melting temperature (Tm) of P (SCL-*co*-MCL 3HA) are reduced in comparison with the traditional PHA polymers, such as P (3HB) or PHBV copolymers. As a result this type of copolymer is able to be processed efficiently and in a cost-effective way. The complete biodegradation can be achieved in either aerobic or anaerobic environments.

Physicochemical properties of P (3HB-*co*-LA) including the molecular weight, thermal properties, and melt viscosity were taken into account as well. The mole fraction of LA monomer has been demonstrated to have an inverse relationship with the molecular weight and the crystallinity of P (3HB-*co*-LA), and direct relationship with the glass transition temperature (Tg) of the polymer [16]. Tg was largely affected by the LA fraction while the melting temperature was only slightly modified, remaining at around 160ºC. Copolymer showed a decreased molecular weight ($M_n$) of 29,000 compared to that of PHA which had a molecular weight of 126,000 [25]. The copolymer exhibited more favourable thermal behavior as well as glass and crystallization transition. Overall the copolymer P (3HB-*co*-LA) had improved mechanical

properties, such as lower viscosity and dynamic moduli (which PLA alone does not possess). A different copolymer, produced by blending MCL PHA with PLA using a melt-mixing method, showed even more improved properties, such as increased toughness, ductility and optical clarity [26].

We have recently reported the expression of the two codon-optimized engineered genes encoding propionate CoA transferase (*pct532*) and PHA synthase (*phaC1400*) which were integrated into *S. meliloti* chromosome [27]. In an attempt to broaden the range of novel polymer types that could be produced, we have now introduced these engineered genes into a broad-host-range plasmid, and expressed them in the two PHA-producing platforms *S. meliloti* and *P. putida*, which exhibit differences in the range and type of PHA that they are able to produce naturally.

## Material and methods

### Bacterial strains and cell culture

All strains used in this study are indicated in Table 1. Strains SmUW499 and SmUW501 were provided by R. Nordeste (University of Waterloo), who constructed them as recently described [28] followed by introduction of *exoF*::Tn5 by transduction from Rm7055. *Escherichia coli* and *Pseudomonas putida* were cultured in Luria-Bertani (LB), while *Sinorhizobium meliloti* was cultured in Tryptone Yeast (TY), supplemented with tetracycline (10 μg ml$^{-1}$) as necessary for plasmid maintenance. Rapid screening of polymer production was performed by adding Nile

**Table 1. Bacterial strains used in this study.**

| Strains | Genotype | Reference |
|---|---|---|
| *Escherichia coli* | | |
| DH5α | *supE44 lacU169(w80lacZDM15) hsdR17 recA1 endA1 gyrA96 thi-1 relA1* | Lab collection |
| DH5α(pRK600) | Helper strain harboring plasmid pRK600 | [31] |
| *Sinorhizobium meliloti* | | |
| Rm1021 | Spontaneous Sm resistant isolate of *S. meliloti* SU47 | [32] |
| Rm7055 | Rm1021 *exoF*::Tn5 | [33] |
| SmUW499 | Rm1021 Δ*phbC exoF*::Tn5 | Ricardo Nordeste |
| SmUW501 | Rm1021 Δ*phbAB* Δ*phbC exoF*::Tn5 | Ricardo Nordeste |
| SmUW255 | SmUW499 harboring pTH1227 | This study |
| SmUW256 | SmUW499 harboring pTAM | This study |
| SmUW257 | Rm7055 harboring pTH1227 | This study |
| SmUW558 | Rm1021 Δ*phbAB* Δ*phbC exoF*::Tn5 harboring pTH1227 | This study |
| SmUW559 | Rm1021 Δ*phbAB* Δ*phbC exoF*::Tn5 harboring pTAM | This study |
| *Pseudomonas putida* | | |
| KT2440 | Wild type (ATCC 47054) | |
| PPUW1 | A spontaneous Rif$^R$ mutant of KT2440 | [17] |
| PPUW2 | Δ*phaC1-phaZ-phaC2*/ ΩKm in PPUW1(Km$^r$) | [17] |
| PPUW18 | PPUW2 harboring pTH1227 | This study |
| PPUW19 | PPUW2 harboring pTAM | This study |
| PPUW20 | PPUW1 harboring pTH1227 | This study |
| PPUW21 | PPUW1 harboring pTAM | This study |

Red (0.5 µg/mL) to YM agar plate, and observing its mucoid phenotype or visualizing by fluorescence [29]. Media formulations were as previously described [30].

For polymer production in *S. meliloti*, strains were initially inoculated in TY, and then 1% overnight culture was transferred to YM media in flasks on a shaker at 180 rpm, 30˚C for 3 days [34,35]. IPTG at 0.4 mM was added into the culture for induction as described.

For polymer production in *P. putida*, strains were initially inoculated in LB, and then 1% overnight culture was transferred to defined media (0.5X E2) [36] in flasks on a shaker at 180 rpm, 30˚C for 3 days [37]. Nile Red (0.5 µg/mL) was added to LB or 0.5X E2 agar plate for rapid screening of polymer production. Sodium octanoate (0.5% w/v) was added in the media when needed.

### β-glucuronidase activity assay

β-glucuronidase (GusA) levels were determined as an indication of the transcript level of synthesized genes in *S. meliloti* strains. The assay was carried out as previously described [37,38] by adding culture into assay buffer at the ratio of 1:4, and incubation at room temperature until the yellow colour developed, at which time the reaction was terminated by the addition of sodium carbonate. Then the absorbance of reaction mixture was measured at 420 nm. The OD at 600 nm of the culture was also recorded for normalization.

### GC analysis

Intracellular polymer production was evaluated by gas chromatography following a protocol which has been described in previous studies [4,39]. Briefly, cells were harvested from flask culture after a 3-day incubation by centrifuging at 4,000 g for 20 min, then washed twice with distilled water, and finally dried at 100˚C overnight. The dried cell weight (DCW) was recorded before methanolysis in 2 ml chloroform and 1 ml PHA solution containing 8 g benzoic acid l$^{-1}$ as an internal standard and 30% sulfuric acid in methanol. The reaction was carried out at 96˚C for 6 h, cooled, and then 1 ml of water was added, the mixture was vortexed, and the solution was allowed to separate into two phases. One µl of the chloroform phase was taken for analysis by GC as previously described [4].

## Results

### Construction of plasmid expressing synthetic codon optimized *pct532* and *phaC1400* genes

The broad host range vector pTH1227 [40], containing the inducible *tac* promoter and *lacI*$^q$ along with a downstream *gusA* gene to use as a reporter, was digested with *Xho*I and *Pst*I, then ligated with the XhoI / PstI -cut synthesized DNA containing the previously described codon optimized sequences [27] encoding the Pct532 and PhaC1400 of Jung et al [4] as shown in Fig 1. This resulted in the plasmid construct pTAM. The pTAM plasmid and the empty vector control pTH1227 were separately introduced into *S. meliloti* and *P. putida* strains by triparental conjugation. Transcription of the introduced genes was confirmed by assay of *gusA*-encoded β-glucuronidase activity.

### Expression of engineered synthesized genes in *S. meliloti*

**a. Phenotypic analysis of strain constructs.** Initial confirmation of polymer production in the constructed strain SmUW256 was performed as previously described for the chromosome engineered strain SmUW254 [27]. The strain was streaked out on agar media alongside *phbC*-positive and *phbC*-negative controls strains (Fig 2). Strain SmUW256, which is the *phbC*

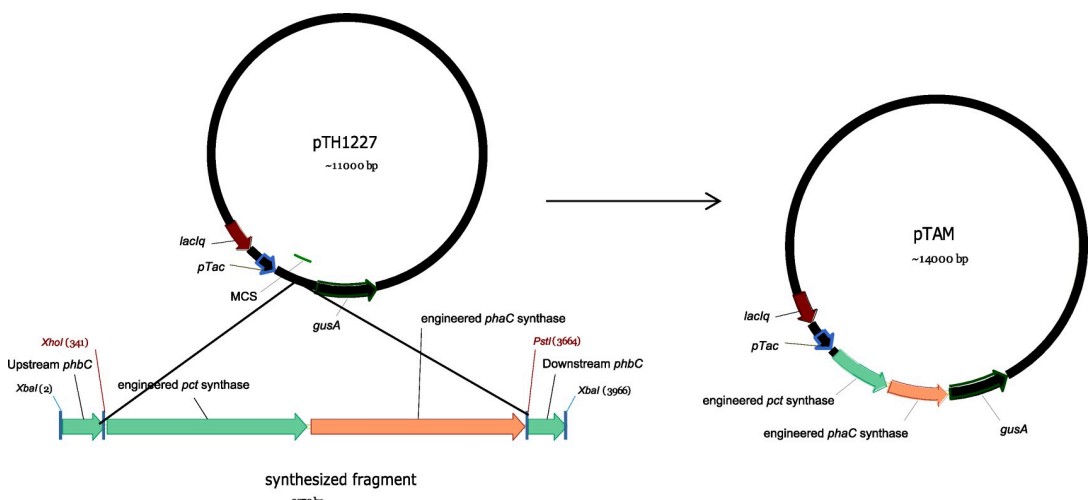

**Fig 1. Construction of expression plasmid pTAM.** Two synthesized genes (*pct532* and *phaC1400*) were inserted as a *Xho*I-*Pst*I fragment into pTH1227 to create pTAM. The synthesized gene sequences have been previously deposited in GenBank and can be accessed via accession Nos. KT382270–KT382273.

mutant containing pTAM, exhibited fluorescence similar to the control strain SmUW257, which contains pTH1227 in a non-mutant *phbC* background. The *phbC* mutant strain SmUW255, containing pTH1227, which cannot produce P (3HB), was deficient in fluorescence, as expected. Interestingly, in the absence of inducer IPTG, we found that SmUW256 exhibited much more fluorescence than with IPTG. This suggested that the gene expression from the plasmid construct was sufficient without IPTG induction to allow complementation of the PHB synthesis deficiency.

**b. Polymer production.** The engineered genes were expressed from pTAM in a *S. meliloti phbC* mutant background under the control of inducible promoter $P_{tac}$ by means of the plasmid-encoded lacI$^q$. Surprisingly, in the absence of IPTG supplementation in YM, levels of

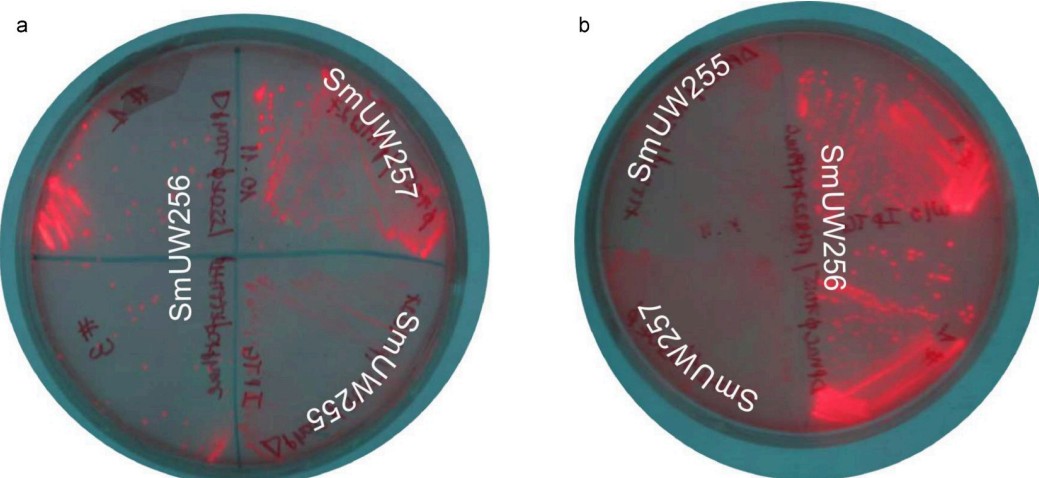

**Fig 2. Phenotypic complementation of plasmid-based *S. meliloti* strains.** a and b demonstrate phenotype difference between plasmid-engineered *S. meliloti* strains and other *S. meliloti* strains: SmUW255 (Δ*phbC exoF*::Tn5 harboring empty plasmid pTH1227), SmUW256 (Δ*phbC exoF*::Tn5 harboring pTAM (#1, 3)), SmUW257 (*exoF*::Tn5 harboring pTH1227). a: induced condition, b: non-induced condition.

PHB produced were similar to the wild-type levels, but in the presence of 0.4 mM IPTG, PHB was only accumulated to about half the wild-type levels. In both conditions, LA was not detected (Fig 3). To investigate whether this was due to issues related to competition between PHB and PLA precursors, we blocked the PHB synthesis pathway by using a *phbAB* mutant background. Feeding the strain with lactic acid (10 g l$^{-1}$) instead of mannitol resulted in PLA homopolymer production of 3.2% DCW (Table 2), demonstrating that in the absence of competing PHB precursors, the PLA precursors could be incorporated into polymer. Higher concentrations of lactic acid resulted in reduced growth, perhaps due to reduction in medium pH.

### c. Investigation of IPTG concentration and induction time

In light of the observation that more polymer was produced in the absence of IPTG than in the presence of IPTG, we decided to further investigate the PHB production under different induction conditions. We also measured GusA expression from the downstream *gusA* gene as a proxy for transcription across the engineered genes. A range of IPTG concentrations from 0 to 1 mM were used to induce gene expression (Fig 4). At an IPTG concentration of 0.05 mM, GusA activity in the SmUW256 strain increased markedly compared to that in the absence of IPTG. However, further addition of IPTG did not substantially increase GusA activity in recombinant strain SmUW256. Meanwhile, the control strain SmUW255 which harbors the empty plasmid pTH1227 showed regulated *gusA* expression which was more proportional to IPTG concentration. Contrary to expectation, increased gene expression did not result in more PHB being produced (Fig 5); however, both DCW and yield decreased relative to increased IPTG concentration (data not shown), hence leading to the overall decreasing levels of PHB. Interestingly, the basal level of expression in absence of IPTG resulted in the best growth and PHB production based on the highest DCW, yield and PHB percentage.

Next, we investigated the effect of induction timing to see if the cells behave differently when 0.4 mM IPTG was added at a later point in time. We observed that the later the IPTG was added into the culture, the lower the GusA activity (Fig 4B). Therefore, if we only consider reporter gene expression level, induction at the beginning of cultivation is the most optimal; however, whether reporter gene expression is directly proportional to PHB production is a separate issue that we sought to resolve. Once again, the results do not show this relationship (Fig 5B). Both DCW and yield were substantially decreased when the induction occurred at the beginning of cultivation, resulting in the lowest PHB percentage. Induction at Day 1 or 2 of cultivation did not make a difference compared to no induction, suggesting that the basal level is still the ideal for PHB production.

### Introduction of engineered synthesized genes into *P. putida*

**a. Phenotypic analysis of strain constructs.** The phenotype of *P. putida* harboring the pTAM and pTH1227 plasmids when grown on LB supplemented with sodium octanoate was examined. Interestingly, we found that the strains carrying the pTAM plasmid showed a distinguishable phenotype from the strains only carrying the empty plasmid pTH1227 (Fig 6). Both the wild type and mutant strains containing the pTAM plasmid showed a milky white colony phenotype, while the strains that carried only the empty plasmid pTH1227 had a yellowish colony color. The milky white colour is likely due to polymer accumulation enabled by pTAM. It appears that the function of the native PHA synthase, and not other enzymes in the *P. putida* PHA biosynthesis pathway, is affected when the strain is cultured in complex media with excess carbon source because the engineered PHA synthase was able to direct the accumulation of PHA on complex media.

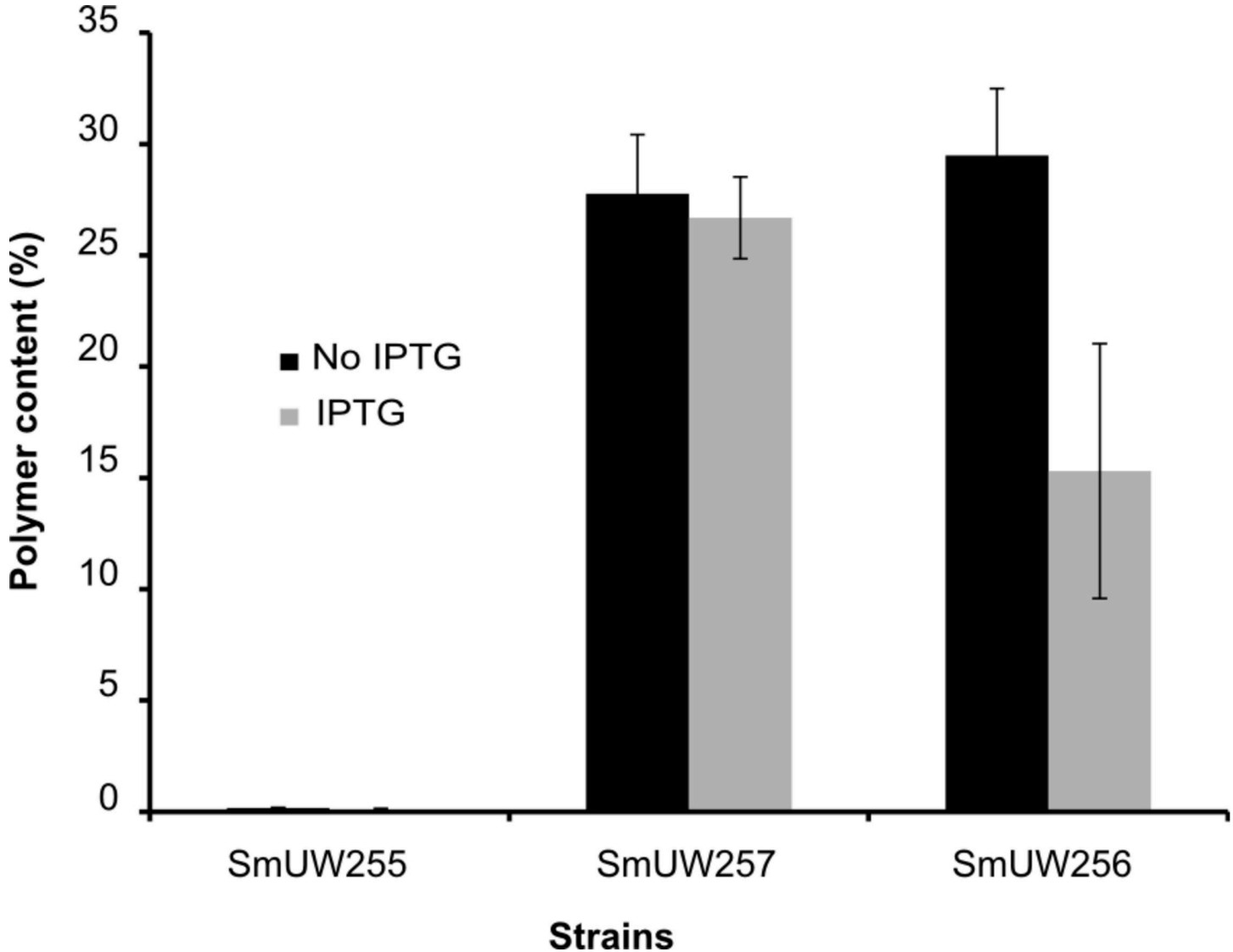

**Fig 3. Polymer content % DCW of SmUW255 (*phbC–*), SmUW257 (*phbC+*) and SmUW256 (engineered *phbC*).** Black bars represent inducing condition (0.4 mM IPTG was added to cultures); grey bars represent non-inducing condition (no IPTG). Experiment was performed in biological duplicate, and error bars represent the range of the mean.

**Table 2. PLA homopolymer production in *S. meliloti phbAB phbC* mutant background harbouring pTAM.**

| | Lactic acid concentration | | | | | |
| | 10 g/l | 20 g/l | 30 g/l | 10 g/l | 20 g/l | 30 g/l |
| Strain | PLA homopolymer content (% DCW) | | | DCW (g/l) | | |
|---|---|---|---|---|---|---|
| SmUW558 (control) | ND | - | - | 0.43 ± 0.03 | - | - |
| SmUW559 | 3.2 ± 1.8 | 0.18 ± 0.13 | ND | 0.35 ± 0.05 | 0.16 ± 0.05 | ND |

Strains were cultured in YM with mannitol substituted by lactic acid at the indicated concentrations.

SmUW558: *ΔphbAB ΔphbC* (pTH1227), SmUW559: *ΔphbAB ΔphbC* (pTAM).—: Not available. ND: Not detectable. The experiment was performed in duplicate.

Results are presented as mean ± standard deviation.

a

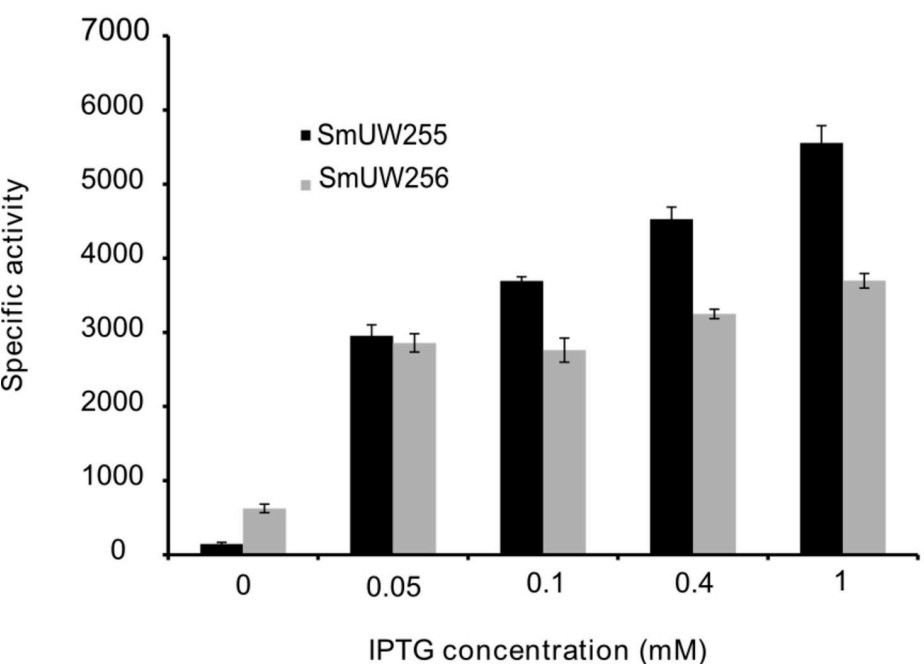

b

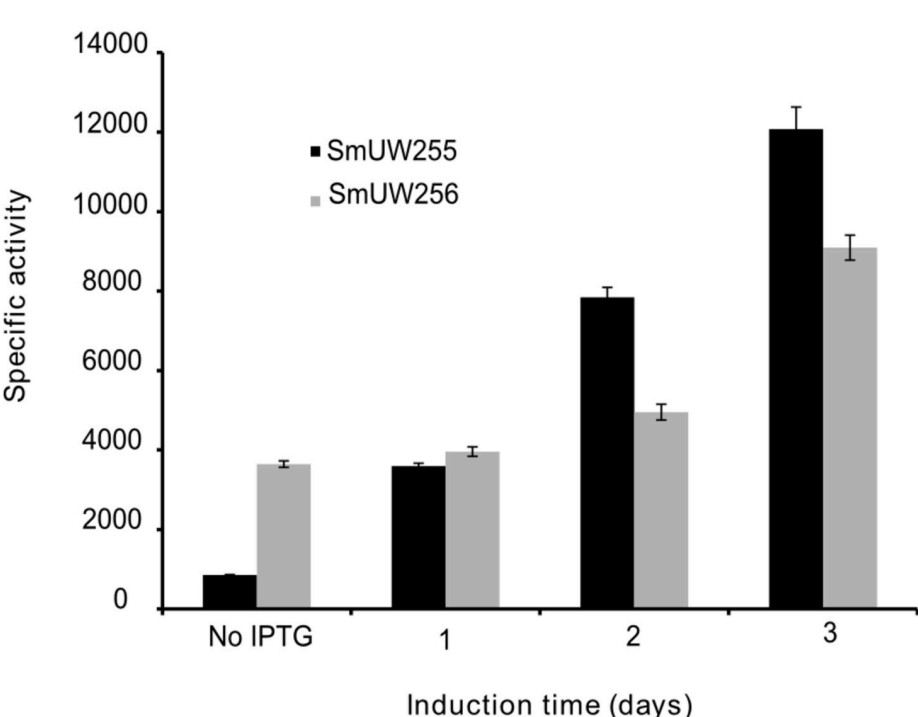

**Fig 4.** Measurement of β-glucuronidase activity of strain SmUW255 (*phbC*), SmUW256 (containing pTAM) induced with different concentrations of IPTG (a) and induced with 0.4 mM IPTG at different points in time for 3 day incubation (b).

We also investigated colony fluorescence on Nile Red containing plates with different growth media (Fig 7). On LB plates without the addition of extra carbon source, no fluorescence was observed for any of strains. Only strains harboring pTAM plasmid showed a strong fluorescence when grown on LB plates with an excess carbon source (eg. sodium octanoate). Meanwhile, the addition of lactic acid did not result in observed fluorescence.

**b. Polymer production in *P. putida* strains growing in defined media supplemented with either sodium octanoate or lactic acid as a substrate.** Strains were cultured in defined media 0.5X E2 supplemented with either sodium octanoate or lactic acid. The negative control strain PPUW18, which is the *P. putida phaC* deletion mutant strain carrying the empty plasmid pTH1227, did not produce detectable polymers on either media. Overall, the media containing sodium octanoate as sole carbon source supported both the growth and the polymer production much better than media with lactic acid as sole carbon source (Table 3). The *phaC* mutant with pTAM plasmid showed the highest DCW and polymer content up to 1.28 g/l and 42%, respectively, using sodium octanoate. The strains carrying the pTAM plasmid were able to incorporate LA monomers to generate a novel quaterpolymer composed of LA, 3HB, 3HHx and 3HO. We found that growth in lactic acid supplied more LA precursors than did sodium octanoate, with the wild-type strain and the mutant strain carrying pTAM accumulating LA of 2.8 and 1.9% mol respectively on lactic acid, while lactic acid was barely detectable on sodium octanoate. Lactic acid is a direct substrate of the modified propionate-CoA transferase, whose function is to add the CoA moiety onto the substrate. Even though lactic acid did not support the cell growth as well as sodium octanoate, it supplied more LA precursor toward novel polymer production. This implies that in *P. putida* very little lactyl-CoA was produced during growth on sodium octanoate substrate.

## Discussion

By using two codon-optimized genes placed in tandem on a broad-host range vector and expressed under an inducible promoter in *S. meliloti* and *P. putida* backgrounds, we were able to demonstrate the production of LA containing polymers. In *S. meliloti*, we demonstrated that this plasmid construct was able to complement the *phbC* mutant strain, but unlike the genome-engineered *S. meliloti* strain which has been shown to be able to produce copolymer P (3HB-*co*-LA) [27], it only produced homopolymer P (3HB). This could be due to insufficient provision of LA precursor. Genetic removal of the ability to produce 3-hydroxybutyryl-CoA substrate for the PHA synthase resulted in production of PLA homopolymer up to 4% when growing in YM supplemented with lactic acid.

The question of how much synthase enzyme protein level is optimal for polymer production is still unresolved. A strategy of maximizing expression does not necessarily translate into higher levels of metabolic end product. For instance, it was reported that *E. coli* XL1-Blue harboring the low-copy-number plasmid pJRDTrcphaCAB$_{Re}$ produced P(3HB) more efficiently than the strain harboring the high copy number pTrcphaCAB$_{Re}$ [41]. There is still not a good understanding of the relationship between synthase gene expression and levels of polymer accumulated. In our study, we have provided an example where the relationship between protein expression and target products is inversely proportional. Increasing the levels of IPTG or lengthening the induction time resulted in production of less polymer.

a)

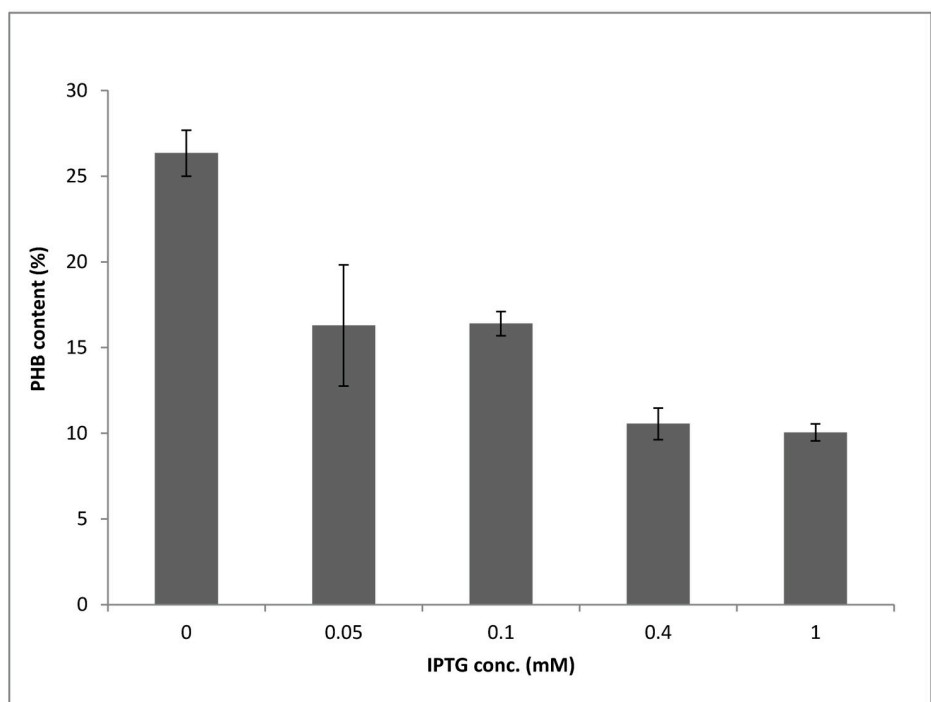

b)

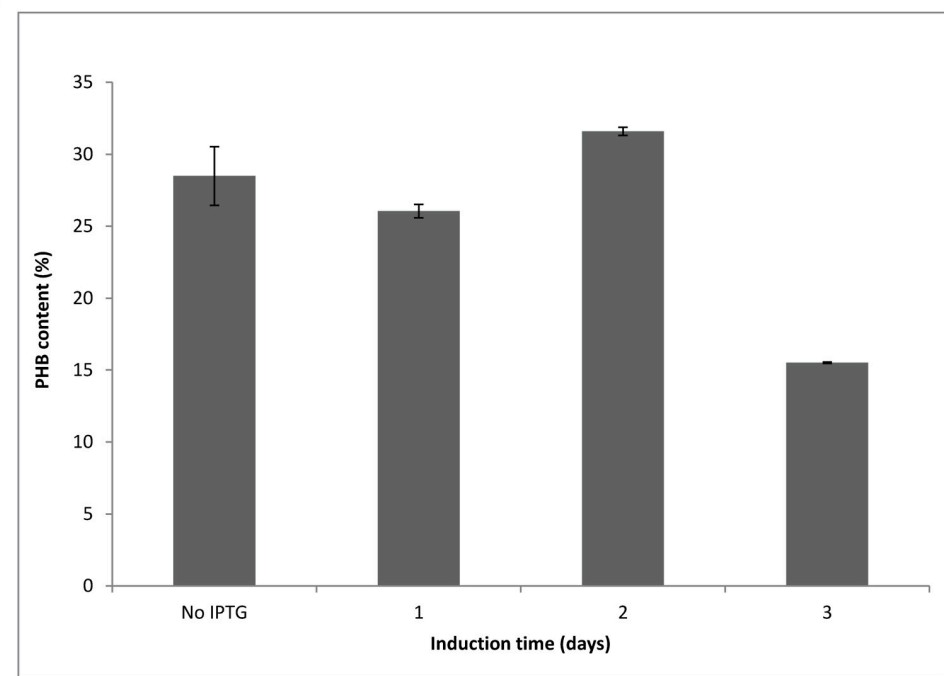

**Fig 5.** PHB content produced by strain SmUW256 induced at different concentrations of IPTG (a) and induced at 0.4 mM IPTG at different points in time for 3 day incubation (b). The experiment was performed in duplicate. Results are presented as mean ± standard deviation (SD). Error bars indicate SD.

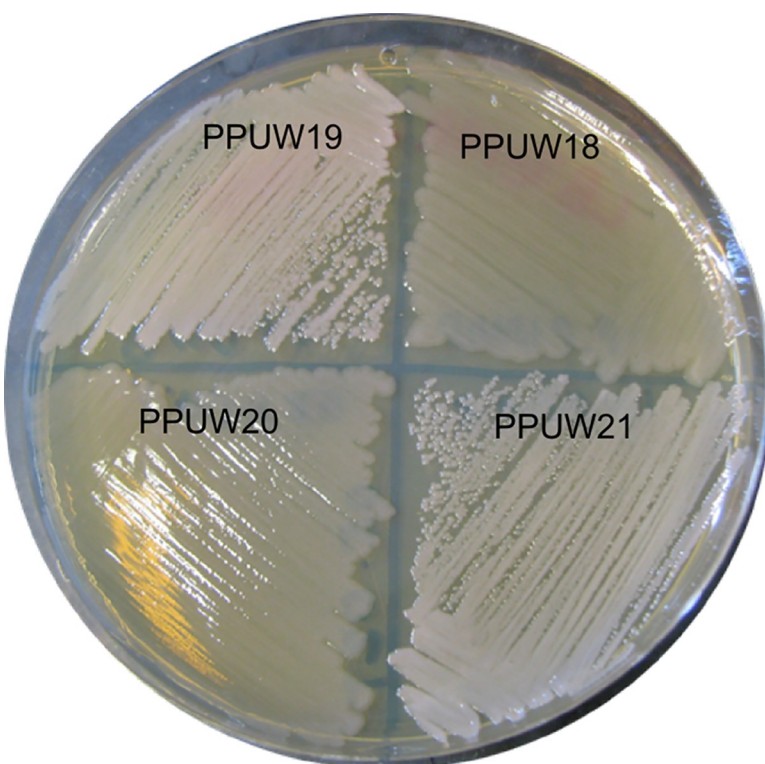

**Fig 6. Phenotype of strains on LB supplemented with sodium octanoate.** A milky white color was observed for PPUW19 (*phaC* mutant harboring pTAM) and PPUW 21 (wild-type strain harboring pTAM). Yellowish color was observed for PPUW18 (*phaC* mutant harboring pTH1227) and PPUW20 (wild-type strain harboring pTH1227).

Upon introduction of pTAM into *P. putida*, the strain was able to produce a novel quater-polymer derived from 4 different monomers (LA, 3HB, 3HHx and 3HO). To our knowledge, a copolymer of this type has never been reported. The production of P(LA-*co*-3HB-*co*-3HHX) was previously demonstrated in *E. coli* via a reverse reaction of the β-oxidation pathway [42]. In that study, the *E. coli* strain was equipped with LA- polymerizing enzyme (LPE) encoding a

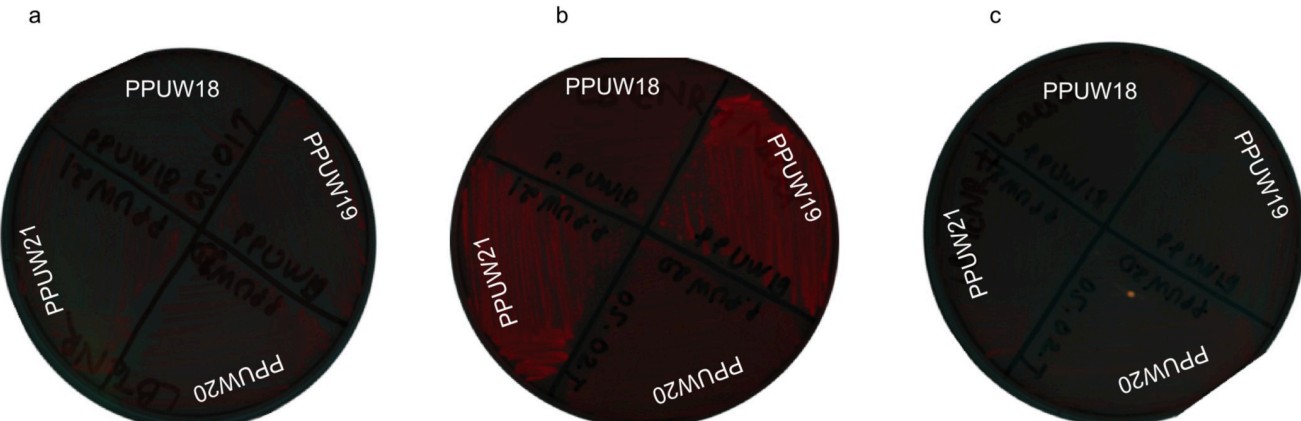

**Fig 7. Fluorescence observation of different *P. putida* strain backgrounds on different PHA and non-PHA accumulating agar plates.** PPUW18 (*phaC* mutant harboring pTH1227), PPUW19 (*phaC* mutant harboring pTAM), PPUW20 (wild-type KT2440 harboring pTH1227), PPUW21 (wild-type KT2440 harboring pTAM). a: LB + Nile Red, b: LB + sodium octanoate + Nile Red, c: LB + lactic acid + Nile Red.

**Table 3. PHA copolymer production in *P. putida* harboring pTAM and control plasmid vector.**

| Strain | Culture Media | | | | | | |
|---|---|---|---|---|---|---|---|
| | **0.5X E2 + Lactic Acid** | | | | | | |
| | LA | 3HB | 3HHx | 3HO | DCW (g/l) | Polymer content (%) | Yield (g/l) |
| PPUW20 | - | 3.3 | 7.3 | 89.4 | 1 ± 0.074 | 5.4 ± 0.178 | 0.057 ± 0.002 |
| PPUW21 | 2.8 | 9.5 | 12.6 | 75.1 | 1 ± 0.163 | 4.9 ± 0.192 | 0.04 ± 0.01 |
| PPUW18 | - | - | - | - | 0.8 ± 0.043 | <0.5 | - |
| PPUW19 | 1.9 | 22 | 10.2 | 65.9 | 0.7 ± 0.088 | 3.4 ± 0.477 | 0.024 ± 0.006 |
| | **0.5X E2 + Sodium Octanoate** | | | | | | |
| | LA | 3HB | 3HHx | 3HO | DCW (g/l) | Polymer content (%) | Yield (g/l) |
| PPUW20 | - | 0.9 | 5.7 | 93.4 | 0.94 ± 0.003 | 26.5 ± 1.5 | 0.25 ± 0.01 |
| PPUW21 | 0.28 | 20.5 | 13.4 | 65.7 | 0.91 ± 0.135 | 35.4 ± 3.3 | 0.32 ± 0.07 |
| PPUW18 | - | - | - | - | 0.72 ± 0.025 | <0.5 | - |
| PPUW19 | 0.4 | 23.9 | 14.2 | 61.4 | 1.28 ± 0.567 | 42 ± 6.1 | 0.53 ± 0.27 |

The experiment was performed in duplicate. Results are presented as mean ± standard deviation.

mutant *phaC1* gene from *Pseudomonas* sp. 61–3, propionyl-CoA transferase (Pct) and (R)-specific enoyl-CoA hydratase (PhaJ4). It is difficult to understand how in that particular study 100% P (3HB) was produced during growth in LB supplemented with glucose since there is no link between the fatty acid synthesis pathway and PHA precursor formation in *E. coli*. In previous studies, *E. coli* was able to produce a small amount of MCL PHA from non-related carbon source only when provided with a PHA polymerase and a modified thioesterase I [43]. In addition, the polymer content of P (LA-*co*-3HB-*co*-3HHX) produced in *E. coli* was extremely low (<5% DCW). In comparison, in our study we observed incorporation of 4 different monomers and an increase of the polymer content up to 42%. In addition, the fraction of LA and 3HB monomers was increased in the complemented mutant PhaC strain to a greater degree than the strain that still contained the wild type *phaC* genes. This is likely due to competition between the polymerase enzymes in the wild-type strain or the higher overall polymerase enzyme activity towards MCL PHAs in the wild-type, resulting in the increase of MCL fraction over LA and 3HB fraction. It has been suggested that the engineered PhaC enzyme has a broader substrate than the native *P. putida* PhaC. The engineered PHA synthase was originally Type II PhaC1 synthase that accepts and polymerizes MCL-3HA (C6-C14) monomers [16]. This enzyme was engineered to broaden the substrate towards SCL-3HAs (specifically, 3HB) and LA. Nonetheless, whether it still retains its ability to accept MCL-3HA has not previously been demonstrated.

## Acknowledgments

We thank Ricardo Nordeste, University of Waterloo, for the gift of strains SmUW499 and SmUW501.

## Author Contributions

**Conceptualization:** Tam T. Tran, Trevor C. Charles.

**Funding acquisition:** Trevor C. Charles.

**Investigation:** Tam T. Tran.

**Methodology:** Tam T. Tran, Trevor C. Charles.

**Supervision:** Trevor C. Charles.

**Writing – original draft:** Tam T. Tran.

**Writing – review & editing:** Tam T. Tran, Trevor C. Charles.

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
