## [Decision Letter · Decision Letter 0]

17 Jul 2019

PONE-D-19-15052

Lactic Acid Containing Polymers Produced in Engineered Sinorhizobium meliloti and Pseudomonas putida

PLOS ONE

Dear %Dr% %Charles%,

Thank you for submitting your manuscript to PLOS ONE. After the requirement of three reviewers for the evaluation, two of them present a strong criticism as it currently stands. Therefore, we invite you to carefully study all their considerations and submit a deeply revised version of the manuscript that addresses the points raised during the review process.

We would appreciate receiving your revised manuscript by Aug 31 2019 11:59PM. To enhance the reproducibility of your results, we recommend that if applicable you deposit your laboratory protocols in protocols.io, where a protocol can be assigned its own identifier (DOI) such that it can be cited independently in the future. For instructions see: http://journals.plos.org/plosone/s/submission-guidelines#loc-laboratory-protocols

We look forward to receiving your revised manuscript.

Kind regards,

Francisco Martinez-Abarca, Ph.D.

Academic Editor

PLOS ONE

Journal Requirements:

1. We note that you have included the phrase “data not shown” in your manuscript. Unfortunately, this does not meet our data sharing requirements. PLOS does not permit references to inaccessible data. We require that authors provide all relevant data within the paper, Supporting Information files, or in an acceptable, public repository. Please add a citation to support this phrase or upload the data that corresponds with these findings to a stable repository (such as Figshare or Dryad) and provide and URLs, DOIs, or accession numbers that may be used to access these data. Or, if the data are not a core part of the research being presented in your study, we ask that you remove the phrase that refers to these data.

Reviewers' comments:

Reviewer's Responses to Questions

**Comments to the Author**

1. Is the manuscript technically sound, and do the data support the conclusions?

Reviewer #1: Partly

Reviewer #2: Yes

Reviewer #3: Partly

2. Has the statistical analysis been performed appropriately and rigorously? 

Reviewer #1: N/A

Reviewer #2: Yes

Reviewer #3: No

3. Have the authors made all data underlying the findings in their manuscript fully available?

Reviewer #1: Yes

Reviewer #2: Yes

Reviewer #3: No

4. Is the manuscript presented in an intelligible fashion and written in standard English?

Reviewer #1: No

Reviewer #2: Yes

Reviewer #3: Yes

5. Review Comments to the Author

Reviewer #1: The manuscript describes the production of lactic acid-containing polymers with medium-chain length units in Sinorhizobium meliloti and Pseudomonas putida. Many cases have been reported in various monomeric constituents using some bacterial platforms starting from 2008. In such a case, this is an example of the series study. Most recently Goto et al. reported the similar paper on the polymer with higher molar-fraction of lactic acid (> 4% in this paper) using Escherichia coli. Thy also provides the polymer properties. On the other hands, any new finding cannot be found in this paper. For example, in several points, new synthetic pathway, higher production titter, enriched lactic fraction, high molecular weight, new polymer properties. The reviewer has to consider this as a not-permitted assessment for publication in the present form.

Reviewer #2: General Comments

This manuscript describes the engineering of Sinorhizobium meliloti and Pseudomonas putida to synthesize novel co-polymers containing both polyhydroxyalkanoates (PHAs) and polylactic acid (PLAs). This work is novel and innovative. The manuscript is well written, the experimental approach and methods are appropriate and well described, and the conclusions are supported by the data. This manuscript will make an excellent contribution to the literature. There are only a few minor issues that need to be addressed before the manuscript may be published.

Specific Comments

Page 6, line 102: In the sentence, “…were provided by R. Nordeste, who constructed them as recently described (27)…”, please provide the institutional affiliation of R. Nordeste.

Page 7, line 115: Please provide the vendor/manufacturer and its location for the culture medium “0.5*E2”

Page 7, lines 120 to 121: In the sentence, “β-glucuronidase levels were determined as an indication of the transcript level of synthesized genes in both E. coli and S. meliloti strains.” Firstly, do the authors really mean E. coli here? Should this not be P. putida?

Second, the assay described is protein (gene product) based. The authors cite two references, one which describes the use of the β-glucuronidase (GusA) assay in actinomycetes (reference 36) and the other describes the use of this assay as a measure of promoter activity in Lactic Acid bacteria (reference 37). In the current paper, the authors suggest that the assay is a direct indicator of the level of transcription of the genes of interest, but this is based on the assumption that the level of protein activity detected is proportional to the level of translation, and this in turn is proportional to the level of transcription. Is this actually the case? Can the authors provide some additional clarification about the relationship between transcription, translation, and enzyme activity in S. meliloti and P. putida?

Figure 3: The resolution of the image for Figure 3 is very poor, and the bands in each lane have merge with each other so it is difficult to tell which lane is which. Can the authors replace this gel image with a better one?

Page 11, lines 202 to 209: The authors refer to GusA, which is the gene product β-glucuronidase. For clarity, please revised the sentence on page 7, line 120, as follows: “β-glucuronidase (GusA) levels were determined….

Reviewer #3: The authors present a plasmid containing the genes encoding the propionate CoA transferase of C. propionicum and a modified PHA synthase I of Pseudomonas sp. MBEL 6-19. They introduced this into phaC mutants of S. melioti and P. putida and report the production of PLA homopolymer in S. meloti and of a novel quaterpolymer P(3HB-co-LA-co-3HH-co-3HO) in P. putida in the presence of octanoate. Though the production of a novel quaterpolymer is potentially interesting for applied processes, the manuscript, in its present state, has a number of shortcomings.

p. 9, ll. 147-152: It would be helpful to have some more information on the genetic arrangement of the plasmid, e.g. how does the gusA reporter work, is it a transcriptional or translational fusion? How was the codon optimization achieved? What does the 532 or the 1400 stand for in the gene names? What RBSs are used and where were they located?

p. 9, ll. 161-167 and Fig. 2: Why is there such a big difference in the fluorescence of SmUW257 comparing Fig. 2a to b? It should be comparable, as only IPTG is added in a) but this strain carries the empty plasmid. At the same time in 2b) it is hard to see a difference between the negative control (SmUW255) and the positive control (SmUW257). In my opinion this assay is not suitable to draw quantitative conclusions, only qualitative if any, as fluorescence depends on the amount and density of cells, which is hard to equal in solid phase medium. To get a rough estimation on the amount of polymer produced, the authors could repeat this plate experiment ensuring that the same amount of cells is plated or stain liquid cultures with the same OD (better cell count).

p.10, ll. 174-179 and Fig. 3: In MHO nothing can be concluded form this WB. Apart from the fact, that it is hard to distinguish the different lanes, no band is observed in the positive control, and the rest is mainly smear. How did the authors ensure that the same amount of protein is loaded in each lane, this should be stated in the M&M part or a Coomassie stained gel should be shown. Which lanes show the sample without IPTG, which is mentioned in the manuscript?

p. 10, Polymer production (Fig.4): How many independent replicates where analyzed? What do the error bars represent? Is the % polymer referred to CDW?

p. 11, Tab. 2: Why is the PLA production negatively correlated to the lactic acid concentration in the medium? Is that expected? Please comment on this.

Fig. 5 and 6: Information on how many replicates were analyzed and what the error bar represent is missing. It would be more coherent to have in Fig. 5b and 6b the same order of the samples (from day 1 to 3 or vice versa).

p. 12, ll. 227-228: The authors state that the gusA reporter expression is inversely proportional to PHB production. I don’t see a correlation between the expression and the polymer content. The GusA activity, which serves a s a proxy for gene expression, seems to be either on or off. The authors should perform a correlation and include a correlation coefficient to support this statement

p.12, l. 229: Reference is made to S1, however it is not available.

p.12, l. 230: How was the significance determined?

Fig. 7a is redundant with Fig. 7b. and can be omitted from the manuscript.

p.14, l. 265: Did the authors mean sodium octanoate?

Fig 9a: When were the samples taken? Is this result of only one analysis? Please comment on this.

Fig. 9b: How do the authors explain the difference in CDW in PPU20, PPU21, and PPU19? PPU19 reaches more than twice the CDW of the wt strain in LB medium. Why is this not observed with minimal medium? Is that a common phenotype, or just observed in this specific experiment?

Technical comments:

1. Information on media composition is completely missing (YM, TY, 0.5*E2)

2. A list of abbreviations would be helpful (especially for the polymer abbreviations, HB, LA, HH. HV etc.).

6. PLOS authors have the option to publish the peer review history of their article (what does this mean?). If published, this will include your full peer review and any attached files.

Reviewer #1: No

Reviewer #2: No

Reviewer #3: No

---

## [Author Response · Author response to Decision Letter 0]

11 Dec 2019

Revisions for Lactic Acid Containing Polymers Produced in Engineered Sinorhizobium meliloti and Pseudomonas putida 

Comments by Reviewer #1:

The manuscript describes the production of lactic acid-containing polymers with medium-chain length units in Sinorhizobium meliloti and Pseudomonas putida. Many cases have been reported in various monomeric constituents using some bacterial platforms starting from 2008. In such a case, this is an example of the series study. Most recently Goto et al. reported the similar paper on the polymer with higher molar-fraction of lactic acid (> 4% in this paper) using Escherichia coli. Thy also provides the polymer properties. On the other hands, any new finding cannot be found in this paper. For example, in several points, new synthetic pathway, higher production titter, enriched lactic fraction, high molecular weight, new polymer properties. The reviewer has to consider this as a not-permitted assessment for publication in the present form.

Response:

We thank the reviewer for this comment. As stated in the comment, our study is another instance of a continuing effort to seek out new platforms for novel polymer production. There have been many reports using Escherichia coli as a host to produce different types of biopolymer (a given example is the paper of Goto et al. mentioned by the reviewer, and which we have now cited). However, as mentioned in our manuscript, using E. coli as a host has shown some drawbacks, such as lacking native system for polymer precursor production or suffering from the stress of polymer production. Therefore, our study has demonstrated the feasibility to produce these types of polymers in native polymer producers such as Sinorhizobium meliloti and Pseudomonas putida.

Comments by Reviewer #2:

General Comments

This manuscript describes the engineering of Sinorhizobium meliloti and Pseudomonas putida to synthesize novel co-polymers containing both polyhydroxyalkanoates (PHAs) and polylactic acid (PLAs). This work is novel and innovative. The manuscript is well written, the experimental approach and methods are appropriate and well described, and the conclusions are supported by the data. This manuscript will make an excellent contribution to the literature. There are only a few minor issues that need to be addressed before the manuscript may be published.

General response:

We apologize for those minor issues. We have addressed them as described following each comment.

Specific Comments

Page 6, line 102: In the sentence, “…were provided by R. Nordeste, who constructed them as recently described (27)…”, please provide the institutional affiliation of R. Nordeste.

Response: 

R. Nordeste, University of Waterloo. This has been added to the manuscript.

Page 7, line 115: Please provide the vendor/manufacturer and its location for the culture medium “0.5*E2”

Response: 

We have added the reference for 0.5X E2. 

Page 7, lines 120 to 121: In the sentence, “β-glucuronidase levels were determined as an indication of the transcript level of synthesized genes in both E. coli and S. meliloti strains.” Firstly, do the authors really mean E. coli here? Should this not be P. putida?

Second, the assay described is protein (gene product) based. The authors cite two references, one which describes the use of the β-glucuronidase (GusA) assay in actinomycetes (reference 36) and the other describes the use of this assay as a measure of promoter activity in Lactic Acid bacteria (reference 37). In the current paper, the authors suggest that the assay is a direct indicator of the level of transcription of the genes of interest, but this is based on the assumption that the level of protein activity detected is proportional to the level of translation, and this in turn is proportional to the level of transcription. Is this actually the case? Can the authors provide some additional clarification about the relationship between transcription, translation, and enzyme activity in S. meliloti and P. putida?

Response:

We should be more clear about the rationale for this experiment. At first, we aimed to develop the plasmid-based system in our primary short-chain-length PHA producer host S. meliloti. Therefore, we conducted this experiment in S. meliloti only. We added E. coli as a positive control along with our host because the reporter gene was originally obtained from E. coli. 

Regarding the second question, we again performed the assay in E. coli and S. meliloti to determine the transcriptional level of the genes of interest. The assay has been applied to quantitatively analyse the activity of the promoter (in terms of gusA expression), hence inferring the transcription of the genes that are constructed under the same promoter with the gusA gene fusions. This assay has been used in different systems such as plants, mosses, algae, fungi and various bacteria (E. coli, lactic acid bacteria, actinomycetes). We obtained similar results in S. meliloti compared to E. coli (data not shown) which strongly suggested the validity of this assay in our host.

Figure 3: The resolution of the image for Figure 3 is very poor, and the bands in each lane have merge with each other so it is difficult to tell which lane is which. Can the authors replace this gel image with a better one?

Response: 

We have removed this figure, as we agree that the quality is poor, and it does not provide much useful information.

Page 11, lines 202 to 209: The authors refer to GusA, which is the gene product β-glucuronidase. For clarity, please revised the sentence on page 7, line 120, as follows: “β-glucuronidase (GusA) levels were determined….

Response:

We have edited the sentence at the reviewer’s suggestion.

Comments by Reviewer #3:

The authors present a plasmid containing the genes encoding the propionate CoA transferase of C. propionicum and a modified PHA synthase I of Pseudomonas sp. MBEL 6-19. They introduced this into phaC mutants of S. melioti and P. putida and report the production of PLA homopolymer in S. meloti and of a novel quaterpolymer P(3HB-co-LA-co-3HH-co-3HO) in P. putida in the presence of octanoate. Though the production of a novel quaterpolymer is potentially interesting for applied processes, the manuscript, in its present state, has a number of shortcomings.

General response:

We apologize for these shortcomings. We have addressed them as described following each comment.

p. 9, ll. 147-152: It would be helpful to have some more information on the genetic arrangement of the plasmid, e.g. how does the gusA reporter work, is it a transcriptional or translational fusion? How was the codon optimization achieved? What does the 532 or the 1400 stand for in the gene names? What RBSs are used and where were they located?

Response:

Figure 1 describes the genetic arrangement of the plasmid. It is a transcriptional fusion driven by the tac promoter under the regulation of lacIq on the vector; both genes of interest were introduced into the multiple cloning site (MCS) in front of the gusA gene. We have described how we constructed the plasmid in the text. 

We described previously (Tran and Charles 2015) that these genes were codon-optimized for S. meliloti using the codon adaptation tool JCat (Grote et al., 2005). The optimized sequences, with a His tag of six histidine residues at the end of each gene (before the stop codon), and a S. meliloti ribosome binding site (RBS) at the beginning of each gene, were then synthesized on a single fragment, with phaC1400 immediately following pct532. The pct gene which was originally from Clostridium propionicum had an increase of GC content from 40.38% to 53.65% after codon-optimization. Meanwhile, the phaC gene which was originally from P. putida did not show any drastic change of GC content (62.14% compared to initial 58.86%) after codon-optimization since P. putida has GC content similar to S. meliloti. In contrast, the overall number of codons drastically decreased across both genes (for the pct gene, the number of codons decreased from 52 to 37; for the phaC gene, the number of codon decreased from 56 to 39). Both numbers (532 and 1400) are used to be consistent with the gene names used by Jung et al. who originally mutated these genes and then screened them for altered enzyme activities.

Grote A, Hiller K, Scheer M, Munch R, Nortemann B, Hempel DC, Jahn D. 2005. JCat : a novel tool to adapt codon usage of a target gene to its potential expression host. Nucleic Acids Res. 33:526–531.

Jung YK, Kim TY, Park SJ, Lee SY. Metabolic engineering of Escherichia coli for the production of polylactic acid and its copolymers. Biotechnol Bioeng [Internet]. 2010;105(1):161–71.

p. 9, ll. 161-167 and Fig. 2: Why is there such a big difference in the fluorescence of SmUW257 comparing Fig. 2a to b? It should be comparable, as only IPTG is added in a) but this strain carries the empty plasmid. At the same time in 2b) it is hard to see a difference between the negative control (SmUW255) and the positive control (SmUW257). In my opinion this assay is not suitable to draw quantitative conclusions, only qualitative if any, as fluorescence depends on the amount and density of cells, which is hard to equal in solid phase medium. To get a rough estimation on the amount of polymer produced, the authors could repeat this plate experiment ensuring that the same amount of cells is plated or stain liquid cultures with the same OD (better cell count).

Response:

We also agree with the reviewer that this assay should only be used to qualitatively determine polymer production in bacteria. We used this technique as our initial rapid screening for assessment of polymer production in the different strains. For quantitative analysis, we used Gas Chromatography method to determine how much polymer was produced, as well as to identify polymer composition.

p.10, ll. 174-179 and Fig. 3: In MHO nothing can be concluded form this WB. Apart from the fact, that it is hard to distinguish the different lanes, no band is observed in the positive control, and the rest is mainly smear. How did the authors ensure that the same amount of protein is loaded in each lane, this should be stated in the M&M part or a Coomassie stained gel should be shown. Which lanes show the sample without IPTG, which is mentioned in the manuscript?

Response:

We have removed the figure.

p. 10, Polymer production (Fig.4): How many independent replicates where analyzed? What do the error bars represent? Is the % polymer referred to CDW?

Response:

The experiment was done in biological duplicate. Error bars represent the range of the mean. The % polymer content is referred to DCW.

p. 11, Tab. 2: Why is the PLA production negatively correlated to the lactic acid concentration in the medium? Is that expected? Please comment on this.

Response:

Even though lactic acid is theoretically an ideal substrate to produce polymer precursor, it does not seem to be a good substrate to support cell growth. Also, the more lactic acid we added, the more adverse effects on growth were observed, perhaps due to lowering pH in the medium. Evidently, we observed a significant decrease in DCW, as shown in Table 2. As a consequence, the overall PLA production is inversely proportional to the lactic acid concentration.

Fig. 5 and 6: Information on how many replicates were analyzed and what the error bar represent is missing. It would be more coherent to have in Fig. 5b and 6b the same order of the samples (from day 1 to 3 or vice versa).

Response:

The figures have been revised, as recommended. The experiment was done in duplicate. Results are presented as mean ± standard deviation (SD). Error bars indicate SD.

p. 12, ll. 227-228: The authors state that the gusA reporter expression is inversely proportional to PHB production. I don’t see a correlation between the expression and the polymer content. The GusA activity, which serves a s a proxy for gene expression, seems to be either on or off. The authors should perform a correlation and include a correlation coefficient to support this statement

Response:

We thank the reviewer for this comment. What we observed in our experiment that the longer we induced the promoter by adding IPTG at different time points, the lower polymer content detected there appeared to be. Meanwhile GusA activity increased in terms of induction time. More importantly, the strains were still able to produce polymer quite efficiently at base level (without induction). It is merely a suggestion that enzyme activity might not always be directly proportional to polymer production. 

p.12, l. 229: Reference is made to S1, however it is not available.

Response:

We have removed this reference to S1

p.12, l. 230: How was the significance determined?

Response:

We have altered the statement to better reflect that we did not determine statistical significance.

Fig. 7a is redundant with Fig. 7b. and can be omitted from the manuscript.

It has been omitted. 

p.14, l. 265: Did the authors mean sodium octanoate?

Response:

It is indeed sodium octanoate.

Fig 9a: When were the samples taken? Is this result of only one analysis? Please comment on this.

Response:

The time of sampling is provided in the Methods section. The experiment was done in duplicate.

Fig. 9b: How do the authors explain the difference in CDW in PPU20, PPU21, and PPU19? PPU19 reaches more than twice the CDW of the wt strain in LB medium. Why is this not observed with minimal medium? Is that a common phenotype, or just observed in this specific experiment?

Response:

We have removed these data from the paper.

Technical comments:

1. Information on media composition is completely missing (YM, TY, 0.5*E2)

Response: 

We have provided references.

2. A list of abbreviations would be helpful (especially for the polymer abbreviations, HB, LA, HH. HV etc.).

Response: 

We have now provided this on first use within the text.

---

## [Decision Letter · Decision Letter 1]

20 Dec 2019

PONE-D-19-15052R1

Lactic Acid Containing Polymers Produced in Engineered Sinorhizobium meliloti and Pseudomonas putida

PLOS ONE

Dear %Dr% %Charles%,

Thank you for submitting your manuscript to PLOS ONE. After careful consideration, we feel that it has merit but does not fully meet PLOS ONE’s publication criteria as it currently stands. Therefore, we invite you to submit a revised version of the manuscript that addresses the points raised during the review process.

We would appreciate receiving your revised manuscript by Feb 03 2020 11:59PM. To enhance the reproducibility of your results, we recommend that if applicable you deposit your laboratory protocols in protocols.io, where a protocol can be assigned its own identifier (DOI) such that it can be cited independently in the future. For instructions see: http://journals.plos.org/plosone/s/submission-guidelines#loc-laboratory-protocols

We look forward to receiving your revised manuscript.

Kind regards,

Francisco Martinez-Abarca, Ph.D.

Academic Editor

PLOS ONE

Reviewers' comments:

Reviewer's Responses to Questions

**Comments to the Author**

1. If the authors have adequately addressed your comments raised in a previous round of review and you feel that this manuscript is now acceptable for publication, you may indicate that here to bypass the “Comments to the Author” section, enter your conflict of interest statement in the “Confidential to Editor” section, and submit your "Accept" recommendation.

Reviewer #2: All comments have been addressed

Reviewer #3: (No Response)

2. Is the manuscript technically sound, and do the data support the conclusions?

Reviewer #2: Yes

Reviewer #3: Yes

3. Has the statistical analysis been performed appropriately and rigorously? 

Reviewer #2: Yes

Reviewer #3: N/A

4. Have the authors made all data underlying the findings in their manuscript fully available?

Reviewer #2: Yes

Reviewer #3: Yes

5. Is the manuscript presented in an intelligible fashion and written in standard English?

Reviewer #2: Yes

Reviewer #3: Yes

6. Review Comments to the Author

Reviewer #2: The authors have adequately address the concerns of this reviewer, and is now acceptable for publication.

Reviewer #3: The manuscript in its present state is more a short communication than a research article. In their revised version the authors have deleted all controversial data, and replaced part of figure 9 with table 3. In the first part the authors show that an already published pathway for polymer production works plasmid-based in S. meliloti, however there is no benefit over the wildtype strain neither in terms of polymer content nor in composition. Only when the PHB pathway is blocked, small amounts of PLA are detectable. Furthermore, there is the unexplained paradox of induction of transcription with IPTG and polymer accumulation. It might be due to metabolic burden, simultaneous degradation of the polymer, or a disbalance of the enzymatic machinery. The authors did not analyze this in more detail. Growth curves and rates could give a hint on this. In my opinion, this first part of the manuscript is marginally innovative and does not contribute significantly to the field.

In the second part, the authors show that by implementing this pathway in P. putida a novel co-polymer can be produced up to around 40% of the CDW. This is new and interesting and a good starting point for further investigations.

Some issues should be corrected before acceptance:

1) Material and methods need to be adapted to the shortens version of this article, e.g. no data is shown for P. putida grown in liquid LB medium for polymer production, however this is still part of this section.

2) The number of analyzed replicates should be stated, either in the figure or table legends or in the Material and methods section.

3) p. 11 l. 195/196: how can there be error bars in a table? Delete this sentence.

4) p.15 l. 238 and table 3: see above and numbers are not presented with +/- standard deviation. Why do the authors use a different nomenclature for the P. putida strains in this table? Or are these other strains?

5) p. 17 ll. 314-320: This sentence is hard to understand as it is very long.

7. PLOS authors have the option to publish the peer review history of their article (what does this mean?). If published, this will include your full peer review and any attached files.

Reviewer #2: No

Reviewer #3: No

---

## [Author Response · Author response to Decision Letter 1]

17 Feb 2020

We have addressed Reviewer 3’s concerns as follows. 

Some issues should be corrected before acceptance:

1) Material and methods need to be adapted to the shortens version of this article, e.g. no data is shown for P. putida grown in liquid LB medium for polymer production, however this is still part of this section

We have removed the information as appropriate. 

2) The number of analyzed replicates should be stated, either in the figure or table legends or in the Material and methods section

This information has been added where it had been missing. 

3) p. 11 l. 195/196: how can there be error bars in a table? Delete this sentence

This has been corrected.

4) p.15 l. 238 and table 3: see above and numbers are not presented with +/- standard deviation. Why do the authors use a different nomenclature for the P. putida strains in this table? Or are these other strains

We have added standard deviation values and changed the strain nomenclature to be more consistent. 

5) p. 17 ll. 314-320: This sentence is hard to understand as it is very long.

 We have revised the sentence structure.

---

## [Editor Report · Decision Letter 2]

19 Feb 2020

Lactic Acid Containing Polymers Produced in Engineered Sinorhizobium meliloti and Pseudomonas putida

PONE-D-19-15052R2

Dear Dr. Charles,

We are pleased to inform you that your manuscript has been judged scientifically suitable for publication and will be formally accepted for publication once it complies with all outstanding technical requirements.

With kind regards,

Francisco Martinez-Abarca, Ph.D.

Academic Editor

PLOS ONE
---

## [Editor Report · Acceptance letter]

28 Feb 2020

PONE-D-19-15052R2 

Lactic Acid Containing Polymers Produced in Engineered *Sinorhizobium meliloti* and *Pseudomonas putida*

Dear Dr. Charles:

I am pleased to inform you that your manuscript has been deemed suitable for publication in PLOS ONE. Congratulations! Your manuscript is now with our production department. 

With kind regards,

on behalf of

Dr. Francisco Martinez-Abarca 

Academic Editor

PLOS ONE